# Effects of Selective Peroxisome Proliferator Activated Receptor Agonists on Corneal Epithelial Wound Healing

**DOI:** 10.3390/ph14020088

**Published:** 2021-01-25

**Authors:** Yutaro Tobita, Takeshi Arima, Yuji Nakano, Masaaki Uchiyama, Akira Shimizu, Hiroshi Takahashi

**Affiliations:** 1Department of Ophthalmology, Nippon Medical School, Bunkyo-ku, Tokyo 113-8603, Japan; y-tobita@nms.ac.jp (Y.T.); n-yuji@nms.ac.jp (Y.N.); uchiyama@nms.ac.jp (M.U.); tash@nms.ac.jp (H.T.); 2Department of Analytic Human Pathology, Nippon Medical School, Bunkyo-ku, Tokyo 113-8603, Japan; ashimizu@nms.ac.jp

**Keywords:** corneal epithelial wound healing, PPAR, alkali burn

## Abstract

The effects of each subtype-selective peroxisome proliferator activated receptor (PPAR) agonist (α, β/δ, γ) on corneal epithelial wound healing were investigated using a rat corneal alkali burn model. After the alkali burn, each PPAR agonist or vehicle ophthalmic solution was instilled topically onto the rat’s cornea. Corneal epithelial healing processes were evaluated by fluorescein staining. Pathological analyses and real-time reverse transcription polymerase chain reactions were performed to evaluate Ki67 (proliferative maker) expression and inflammatory findings. The area of the corneal epithelial defect at 12 h and 24 h after the alkali burn was significantly smaller in each PPAR group than in the vehicle group. Ki67 mRNA expression was increased in the PPARβ/δ group, whereas mRNA expressions of inflammatory cytokines were suppressed in all of the PPAR agonist groups. Nuclear factor kappa B (NF-κB) was the most suppressed in the PPARγ group. The accelerated corneal epithelial healing effects of each PPAR ligand were thought to be related to the promotion of proliferative capacity and inhibition of inflammation.

## 1. Introduction

The peroxisome proliferator-activated receptors (PPARs) are a group of nuclear receptors belonging to the steroid hormone receptor superfamily [1,2,3]. PPARs consist of three main subtypes: α, β/δ, and γ [1]. The PPARs are involved in glucose and lipid metabolism in humans [1,2]. PPAR agonists play important roles in adipocyte differentiation and lipid metabolism [4]. Whereas PPARα and PPARγ agonists are widely recognized as drugs for dyslipidemia and diabetes [5,6], several studies have reported that PPARs not only have roles in the transcriptional regulation of metabolism, but also in inflammation, angiogenesis, and fibrotic reactions [7,8]. Activation of all subtypes of PPARs has been reported to suppress inflammation via inhibition of NF-κB [9,10]. In the field of ophthalmology, we previously reported the anti-inflammatory and anti-neovascular effects of PPARα and PPARγ in a corneal wound model [11,12,13]. On the other hand, our recent study showed that PPARβ/δ promotes neovascularization while suppressing inflammation [14]. We reported that each subtype of PPAR is localized differently in the eye [15]. Therefore, differences in their localization are thought to be involved in PPAR functions. In terms of localization, we found that all subtypes of PPARs are present in the corneal epithelium [15], and PPARα and β/δ have been reported to play an important role in skin wounds [16,17]. Thus, in the present study, the effects of each subtype of PPAR agonist on corneal epithelial wound healing were examined in a rat alkali burn model.

## 2. Results

### 2.1. Corneal Epithelial Wound Healing

Corneal epithelial defects were created by an alkali burn, and the vehicle ophthalmic solution or one of each PPAR agonist (PPARα: 0.05% fenofibrate, PPARβ/δ: 0.05% GW501516, PPARγ: 0.1% pioglitazone) was instilled onto the cornea immediately after injury and then every 6 h. Real-time RT-PCR showed upregulation of each PPAR by instillation of corresponding agonists (Figure 1a–c), suggesting that the expressions of PPARs were ligand-dependently increased. The corneal epithelial defects after the alkali burn were observed consecutively by fluorescein staining (Figure 2a). The ratio of the epithelial defected area to the original defected area at each time point was significantly lower for the PPAR treatments after 12 h (*p* < 0.05) and even more pronounced at 24 h (*p* < 0.01) (Figure 2b). The ratios of the defected area at 12 h after injury were as follows: PPARα group (30.4% ± 11.5%), PPARβ/δ group (22.2% ± 9.2%), PPARγ group (25.5% ± 9.1%), and vehicle group (53.1% ± 3.3%). The ratios of corneal epithelial defects 24 h after injury were as follows: PPARα group (11.7% ± 5.6%), PPARβ/δ group (5.0% ± 6.5%), PPARγ group (8.0% ± 7.7%), and vehicle group (39.5% ± 20.5%).

### 2.2. Ki67 Expression

Next, Ki67 expression was investigated to evaluate the proliferative potential of corneal epithelial cells. Since Ki67 and inflammation described later were investigated as mechanisms for promoting corneal epithelial repair, their evaluations were performed 6 h after injury, earlier than 12 h, when there was a significant difference (Figure 1). On immunostaining, Ki67 expressions appeared in corneal epithelium of all groups (Figure 3a–d). The number of cells stained in the corneal epithelial cells was significantly higher in the PPARβ/δ group than in the vehicle group (Figure 3e). Other PPAR groups were not significantly different from the vehicle group. Real-time RT-PCR showed a significantly greater increase of Ki67 mRNA in the PPARβ/δ group than in the other groups (Figure 3f). 

### 2.3. Nuclear Factor Kappa B (NF-κB) and Kappa Light Polypeptide Gene Enhancer in the B-Cell Inhibitor, Alpha (I-kBα) Expression

On immunostaining, NF-κB-positive inflammatory cells were observed at the corneal limbus 6 h after alkali burn. Each member of the PPAR group showed a smaller degree of inflammatory cell infiltration compared to the vehicle group (Figure 4a–d). The number of cells expressing NF-κB in the nucleus was smaller in each PPAR group than in the vehicle group. Real-time RT-PCR showed significant suppression of mRNA expression of NF-κB in the PPARγ group compared to the vehicle group 6 h after the alkali burn (Figure 4f).

Similarly, the effect of each PPAR agonist on kappa light polypeptide gene enhancer in the B-cell inhibitor alpha (I-kBα) expression, which is an inhibitory protein of NF-κB, was investigated. I-κBα was strongly expressed in the cell nucleus in the PPARα and PPARβ/δ groups 6 h after the alkali burn (Figure 5b,c). The number of I-κBα-positive cells was larger in the PPARα and PPARβ/δ groups than in the vehicle group (Figure 5e). Real-time RT-PCR showed significant upregulation of mRNA expression of I-κBα in the PPARα and PPARβ/δ groups compared to the vehicle group at 6 h (Figure 5f). Double immunofluorescence studies demonstrated that PPARα and PPARβ/δ were expressed in I-κBα-positive cells (Figure 5g), suggesting that PPARα and PPARβ/δ expressions were associated with the upregulation of I-κBα. There was no correlation between I-kBα and PPARγ (data not shown). These results showed a difference in the involvement of NF-κB and I-κBα in PPARα, PPARβ/δ, and PPARγ functions.

### 2.4. Inflammatory Cytokines

Real-time RT-PCR was performed to compare the expressions of inflammatory cytokines, including TNF-α, IL-1β, and IL-6, 6 h after the alkali injury. All PPAR treatments suppressed the expression of TNF-α, IL-1β, and IL-6 (Figure 6a–c).

## 3. Discussion

In ophthalmology, PPARs have recently been reported to affect inflammation, fibrosis, and angiogenesis [13,14,18,19]. However, there are few reports of the effects of PPARs on corneal epithelial wound healing. The involvement of PPARs in wound healing has often been reported in the field of dermatology [16]. A previous study reported that PPARα and PPARβ/δ expressions were upregulated during the repair process, whereas PPARγ remained undetectable in the wounded murine interfollicular epidermis [20]. PPARα activation was reported to induce skin healing via modulation of the inflammatory phase, and PPARβ/δ activation was reported to protect the wound edge keratinocytes from the TNF-a-induced apoptosis [17,21]. In addition, it was reported that wound healing was delayed in mutant PPARα and PPARβ/δ mice [22]. In this study, the effects of each subtype of PPAR agonist on corneal epithelial wound healing were investigated using a rat corneal alkali burn model. Healing of epithelial defects was promoted in all PPAR groups compared to the vehicle group.

In the present study, PPARβ/δ ligand increased the expression of Ki67, suggesting that activation of PPARβ/δ promotes wound healing by improving proliferative ability. However, contrary to the present results, Gu et al. reported that a PPARβ/δ agonist suppressed Ki67, and a PPARβ/δ antagonist promoted Ki67 in the rat phototherapeutic keratectomy model [23]. Since there are few reports of the involvement of PPARβ/δ in corneal proliferative capacity, further studies using various wound models are needed.

NF-κB is a key regulator of immune development, immune responses, inflammation, and cancer [24,25,26]. When inactivated, it is localized to the cytoplasm by I-κBα, the suppressor protein that binds to NF-κB [24]. I-κBα inhibits NF-kB nuclear translocation and restricts transcription downstream of the NF-κB signaling pathway [24]. Previous studies have reported that PPARα and PPARβ/δ suppress NF-κB [9,11,14]. In the present research, immunostaining and cell counting analyses showed that all PPAR subtypes suppressed NF-κB expression. On the other hand, real-time RT-PCR analysis showed that only PPARγ suppressed mRNA expression of NF-κB. The counts of cells that were NF-κB-positive only in the nuclei were compared. In contrast, RT-PCR analysis measured the total expressions of NF-κB mRNA in the cornea. From this result, it was thought that PPARα and PPARβ/δ suppressed nuclear translocation; NF-κB was expressed in the cytoplasm. Therefore, there appeared to be not much difference in the expressions of NF-κB mRNA between each group compared to the cell count results. The results of I-κBα expression support this. Administration of the PPARγ agonist did not promote the activation of I-κBα, suggesting that the PPARγ agonist suppresses NF-κB without involving the I-κBα pathway. Interestingly, in that regard, it has been reported that PPARγ showed an anti-inflammatory effect by inducing M2 macrophages that suppress inflammation [27]. On the other hand, immunostaining showed that PPARα and PPARβ/δ increased I-κBα expression. Double immunofluorescence staining showed that PPARα and PPARβ/δ were strongly expressed in I-κBα-positive cells. In addition, there was a significant upregulation of mRNA expression of I-κBα in the PPARα and PPARβ/δ groups compared to the vehicle group. These results suggest that PPARα and PPARβ/δ agonists suppress inflammation by inhibiting translocation of NF-κB into the nucleus via the upregulation of I-κBα. The mRNA levels of other inflammatory cytokines (TNF-α, IL-1β, and IL-6) were suppressed in all PPAR groups. It has been reported that all PPAR subtypes have anti-inflammatory effects [9,10]. The present results were similar to those previously reported. Nakamura et al. reported that PPARβ/δ ligand promoted corneal epithelial wound healing, and they suggested that the mechanism may involve suppression of corneal epithelial cell death due to inflammation [28]. Since an alkali burn induces destructive inflammation in the cornea [29], suppression of inflammatory cell death is thought to contribute strongly to wound healing.

In summary, administration of agonists of all PPAR subtypes promoted corneal epithelial wound healing. Reductions of NFKB and TNF-α indicate anti-inflammatory effects, as observed in all PPAR groups, which may be considered to be a factor promoting wound healing. In addition, the PPARβ/δ agonists may accelerate healing by promoting proliferation. The mechanisms of the anti-inflammatory effects of PPARs seemed to be different for each subtype. There may also be different mechanisms of action for each PPAR subtype in the corneal epithelial wound healing process itself. Investigation of the roles of PPARs in the field of ophthalmology has just started, and further research is needed.

## 4. Materials and Methods

### 4.1. Ethics Statement

All animal experiments were conducted in compliance with the Experimental Animal Ethics Review Committee of Nippon Medical School (approval number: 29-055), Tokyo, Japan, and all procedures conformed to the requirements of the Association for Research in Vision and Ophthalmic and Visual Research.

### 4.2. Alkali Burn Model

Eight-week-old, male Wistar rats weighing 200 g were obtained from Sankyo Laboratory Service, Tokyo, Japan. A circular filter paper (diameter, 3.2 mm) that had been soaked in 1 N NaOH was placed on the central cornea for 1 min with the animal under general isoflurane anesthesia to create a corneal alkali burn on the right eye. The left eye remained untreated as a control. The corneas were rinsed with 40 mL of physiological saline after alkali exposure.

### 4.3. Treatment with Each PPAR Agonist

After alkali injury, each ophthalmic solution described below was administered. This study used four kinds of ophthalmic solutions: a vehicle solution, a 0.05% fenofibrate solution (PPARα; Wako Pure Chemical Industries, Osaka, Japan) [11], a 0.05% GW501516 solution (PPARβ/δ; Alexis Biochemicals, Lausanne, Switzerland) [28], and a 0.1% pioglitazone solution (PPARγ; Molekula Ltd., Dorset, UK) [12]. Ophthalmic vehicle solution was prepared as previously described [11,12,15]. One of the ophthalmic solutions was topically instilled onto the ocular surfaces of each rat’s eye. Topical administration was continued in each group immediately after injury and every 6 h until the appropriate endpoint (6 h, 12 h, 24 h after alkali exposure) was reached. Rats reaching each endpoint were euthanized by exsanguination under 3.5% isoflurane anesthesia.

### 4.4. Evaluation of the Corneal Epithelial Defect Area

Corneal epithelial defects in each group were stained with fluorescein solution at 0 h, 12 h, and 24 h after the alkali injury. Then, macroscopic photographs were taken under a blue filter. The green areas in the photographs were considered the corneal epithelium defect areas. The area ratio of the green areas to the entire cornea was calculated using Fiji software (Fiji, ImageJ, Wayne Rasband, National Institutes of Health, Bethesda, MD, USA) [30]. The method for measuring the corneal epithelial defect area using Fiji software was shown in Appendix A.

### 4.5. Histological and Immunohistochemical Analyses

Histological and immunohistochemical analyses were performed as previously described [11,12,15]. Primary antibodies used for the immunohistochemical analyses were: (1) anti-rat Ki67 (Dako Cytomation, Glostrup, Denmark); (2) polyclonal rabbit anti-NF-κB/P65 (Santa Cruz Biotechnology, Dallas, TX, USA); (3) monoclonal mouse anti-I-κBα (Santa Cruz Biotechnology); (4) monoclonal rabbit anti-PPARα (Thermo Scientific, Pierce Biotechnology, IL, USA); and (5) polyclonal rabbit anti-PPARβ/δ (Thermo Scientific). Histofine Simple Stain Rat MAX-PO (Multi, Nichirei Bioscience, Tokyo, Japan) was used as the secondary antibody in both immunostaining procedures. PPARα, PPARβ/δ, and I-kBα were detected by examining frozen tissue sections using double immunofluorescence staining for PPARα (mouse; Texas Red), PPARβ/δ (mouse; Texas Red), or I-kBα (goat; fluorescein isothiocyanate)

### 4.6. Real-Time Reverse Transcription Polymerase Chain Reaction (RT-PCR)

For the RT-PCR analyses, dissected corneal tissues (*n* = 5 for each group at 6 h after corneal injury) were immediately placed into RNAlater solution (Life Technologies, Carlsbad, CA, USA) and stored at −80 °C. Total RNA was extracted from the cornea using an RNeasy FFPE Kit (Qiagen, Hilden, Germany) in accordance with the manufacturer’s protocol. An ND-1000 v3.2.1 spectrophotometer (NanoDrop Technologies, Wilmington, DE, USA) was used to ensure RNA concentration and purity (A260/A280). Libraries of cDNA were created from 4 μg of total RNA using a High-Capacity cDNA Reverse Transcription Kit (Thermo Fisher Scientific) in accordance with the manufacturer’s protocol. Gene expression levels were analyzed using 0.3 μL cDNA with real-time detection of accumulated fluorescence in accordance with the manufacturer’s manual (QuantStudioTM 3 Real-Time PCR System, Thermo Fisher Scientific). Normalized values for mRNA expression in each sample were calculated as the relative quantity of the housekeeping gene, β-actin. Primers used for real-time RT-PCR included: mβ-actin, 5′-GCA GGA GTA CGA TGA GTC CG-3′ (forward) and 5′-ACG CAG CTC AGT AAC AGT CC-3′ (reverse); mPPARα, 5′-TCG TGG AGT CCT GGA ACT GA-3′ (forward) and 5′-GAG TTA CGC CCA AAT GCA CC -3′ (reverse); mPPARβ/δ, 5′-GCC GCC CTA CAA CGA GAT CA -3′ (forward) and 5′-CCA CCA GCA GTC CGT CTT TGT -3′ (reverse); mPPARγ, 5′-GCG AGG GCG ATC TTG ACA -3′ (forward) and 5′- ATG CGG ATG GCC ACC TCT TT-3′ (reverse); mKi67, 5′- ATT TCA GTT CCG CCA ATC C -3′ (forward) and 5′- GGC TTC CGT CTT CAT ACC TAA A -3′ (reverse); mTNF-α, 5′- AAA TGG GCT CCC TCT CAT CAG TTC-3′ (forward) and 5′- TCT GCT TGG TGG TTT GCT ACG AC -3′ (reverse); mIL-1β, 5′-TAC CTA TGT CTT GCC CGT GGA G-3′ (forward) and 5′- ATC ATC CCA CGA GTC ACA GAG G-3′ (reverse); mIL-6, 5′- GTC AAC TCC ATC TGC CCT TCA G A-3′ (forward) and 5′-GGC AGT GGC TGT CAA CAA CAT-3′ (reverse); mNF-κB, 5’-GGCAGCACTCCTTATCAA-3’ (forward) and 5’-GGTGTCGTCCCATCGTAG-3’ (reverse); and mI-κBα, 5’-TGACCATGGAAGTGATTGGTCAG-3’ (forward) and 5’-GATCACAGCCAAGTGGAGTGGA-3’ (reverse). The SDS 2.3 software program (Applied Biosystems) was used for all quantifications.

### 4.7. Statistical Analyses

All results are reported as mean ± standard error. Groups were compared using one-way analysis of variance followed by the Tukey–Kramer post hoc test. (GraphPad Prism, software version 8.4.2, GraphPad Software, San Diego, CA, USA). A value of *p* < 0.05 was considered significant. All analyses were calculated by GraphPad Prism software (Version 8.4.2, GraphPad Software).

## Figures and Tables

**Figure 1 pharmaceuticals-14-00088-f001:**
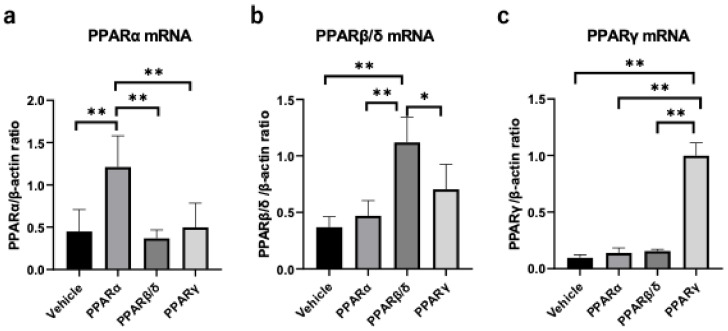
Expression of PPARs. Instillation of each PPAR ligand increased the mRNA levels of the corresponding PPARs in the cornea 6 h after the alkali burn (**a**–**c**). Data are expressed as mean ± standard error (*n* = 8 samples/group). ** *p* < 0.01, * *p* < 0.05.

**Figure 2 pharmaceuticals-14-00088-f002:**
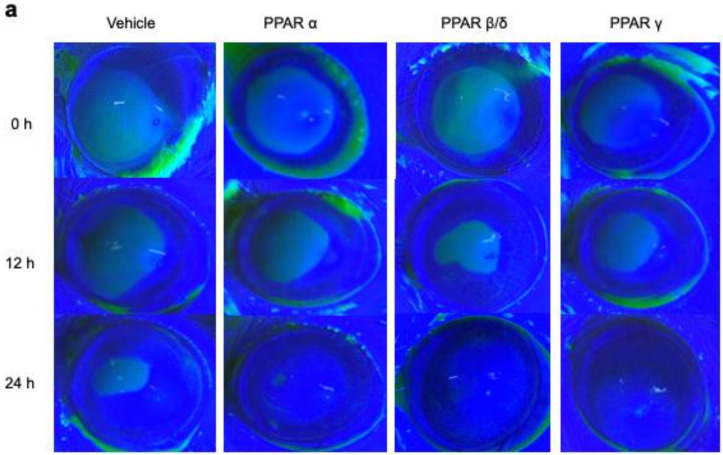
Comparison of corneal epithelial healing processes. (**a**) Representative photographs of rat ocular surfaces 0, 12, and 24 h after alkali injury. Green areas represent fluorescein-stained wounded areas of the ocular surfaces. (**b**) The remaining area of the wound (percent of each initial wounded area) is shown for 12 and 24 h after the alkali burn. At 12 and 24 h after the injury, the reduction rate of the corneal epithelial defect was significantly higher in each PPAR group than in the vehicle group. Data are expressed as mean ± standard error (*n* = 8 samples/group). ** *p* < 0.01, * *p* < 0.05.

**Figure 3 pharmaceuticals-14-00088-f003:**
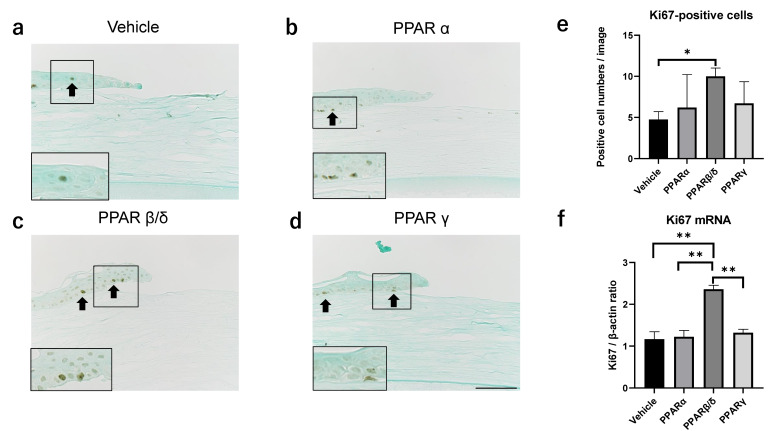
Evaluation of Ki67 expression. (**a**–**d**) Immunostaining of Ki67 in each group 6 h after the alkali burn. (**e**) The number of cells expressing Ki67 in the corneal epithelial cells. The total number of Ki67-positive cells was counted at two locations where the epithelium shown in half of the screen. (**f**) Ki67 mRNA expression 6 h after injury. Higher magnification figures of the boxed area are also shown. Bar, 50 μm. The PPARβ/δ group has significantly more Ki67-positive cells than the vehicle group. Ki67 mRNA expression is significantly higher in the PPARβ/δ group than in the other groups. Data are expressed as mean ± standard error (*n* = 5 samples/group). * *p* < 0.05 or ** *p* < 0.01.

**Figure 4 pharmaceuticals-14-00088-f004:**
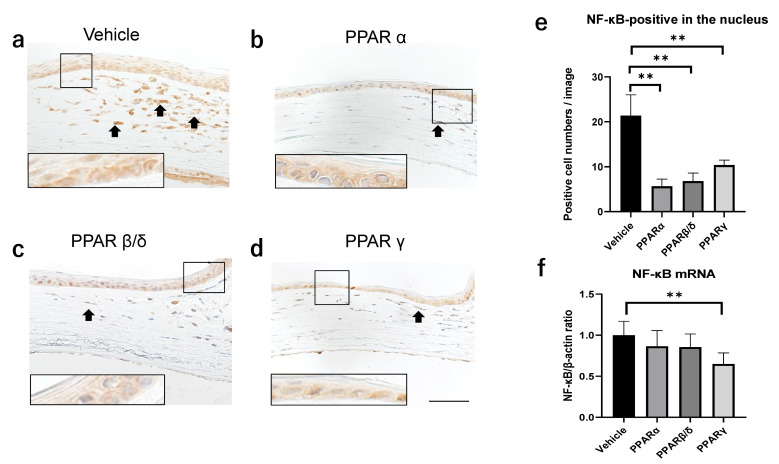
Expression of NF-κB. (**a**–**d**) Immunostaining of NF-κB on the corneal periphery in each group 6 h after the alkali burn. Higher magnification figures of the boxed area are also shown. Bar, 50 μm. (**e**) The number of cells expressing NF-κB in the nucleus. (**f**) NF-κB mRNA expression 6 h after injury. The number of cells stained in the nucleus is significantly lower in each PPAR group than in the vehicle group. NF-κB mRNA expression in the PPARγ group is significantly suppressed compared to the vehicle group. Data are expressed as mean ± standard error (*n* = 8 samples/group). ** *p* < 0.01.

**Figure 5 pharmaceuticals-14-00088-f005:**
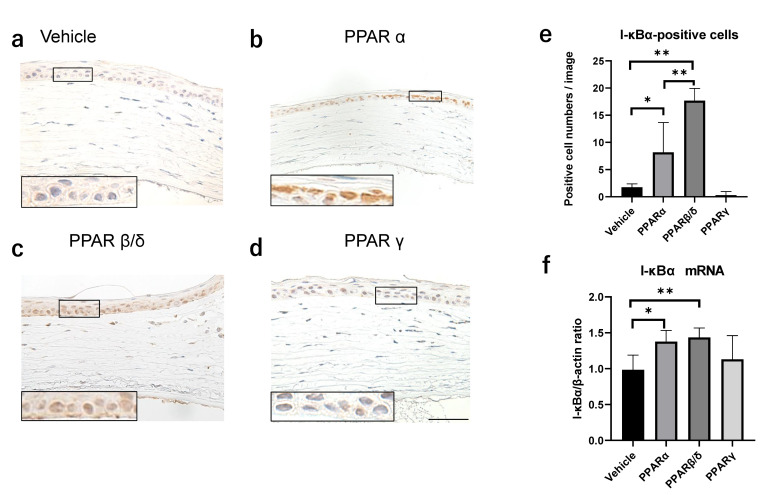
Expression of I-κBα. (**a**–**d**) Immunostaining of I-κBα on the corneal periphery in each group 6 h after the alkali burn. Higher magnification figures of the boxed area are also shown. Bar, 50 μm. (**e**) The number of cells expressing I-κBα in the nucleus. (**f**) I-κBα mRNA expression 6 h after injury. (**g**) Double immunofluorescence studies using I-κBα and each PPAR antibody. Bar, 50 μm. There is a significantly higher number of I-κBα-positive cells in the PPARα and PPARβ/δ groups versus the vehicle group. I-κBα-stained invasive cells coincide with the positively stained PPARα or PPARβ/δ cells (white arrows). Data are expressed as mean ± standard error (*n* = 8 samples/group). * *p* < 0.05 or ** *p* < 0.01.

**Figure 6 pharmaceuticals-14-00088-f006:**
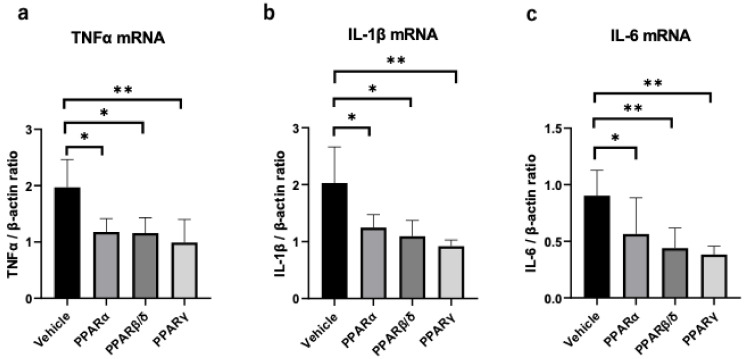
Expression of proinflammatory cytokine mRNAs in the cornea 6 h after alkali injury. The mRNA expression levels of TNF-α (**a**), IL-1β (**b**), and IL-6 (**c**) were measured. Treatment with all PPAR agonists suppressed the mRNA levels of IL-1β, IL-6, and TNF-α. Data are expressed as mean ± standard error (*n* = 8 samples/group). ** *p* < 0.01, * *p* < 0.05.

## Data Availability

Not applicable.

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
