# Peer review of "Effects of Selective Peroxisome Proliferator Activated Receptor Agonists on Corneal Epithelial Wound Healing"

_pharmaceuticals, 2021, doi:10.3390/ph14020088_

Round 1
Reviewer 1 Report
First of all an interesting paper of the role of PPARs on corneal damage healing. Of course, there are a few things to consider:
Line 30: Check the usage of PPAR as multiple proteins or as “one”.
“peroxisome proliferator-activated receptor (PPAR) is a nuclear receptor”
Or is it:
peroxisome proliferator-activated receptors (PPARs) are a group of nuclear receptors
then the second sentence:
“The PPAR has three isoforms” would be: PPARs comprise of three main isoforms.
Note “main” is probably better as splice variants are also isoforms and there are several for each PPAR.
see line 40: subtypes of PPAR s are…
Check this throughout the manuscript.
Line 33: I am sure that PPARa and PPARy are not recognized as drugs! I suppose you mean: PPARa and PPARy <<agonist>>. Please check this. Note sentence is also really long.
Line 34: Add a dot after the “[5.6]” and start a new sentence “Several…..”
Line 37: here you mention “PPARa and y” while before “PPARa and PPARy” was used. Please be consistent and use the PPAR before the y or b/d all the time. (like for example what is done in line 90)
Line 36-39: good that you mention your own research. Its unclear however whether the second inflammatory response was found in the corneal wound model. Please elaborate. Note maybe good to include others as well as PPARs are well known to suppress inflammation.
Line 39-40: “Each PPAR is localized differently in the eye, and differences in their localization are thought to be involved in PPAR function.” References are missing and this info is useless now please elaborate how its then differentially expressed.
Next line you state that they are ALL present in the corneal epithelium. How are the expression levels? Just one reference? Directly isolated or tissue culture? Seems this is a study from one of the authors please mention in that case “we found that all subtypes” line 40.
Line 56-58: “In each…. groups” is bit overdone. remove this.
Just change line 51-52: “The ratio of the epithelial defected area to the original defected area at each time point was significantly lower for the PPAR treatments after 12h (p<0.05) and even more pronounced at 24h (p<0.01) (Figure 1b).”
<<note: this makes the sentence more useful instead of “materials and methods”.>>
Major point:
Figure 2 e: I only see an increase in PPARb/d not for the others! Please then don’t pretend with a “greater increase” as there is no increase for the other PPARs. Change this. Ki67 is unchanged for PPARa and PPARy though interestingly increased for PPARb/d
Also change this in the discussion.
Line 69: for the other two PPARs you could better state: “seemed more enhanced”.
Maybe even better to start with the mRNA as that result is clear and supported by the immunostaining. As normally you present the clearest typical pictures. Therefore, for me only the PPARb/d reflect an increase in protein supporting your mRNA increase the other two might be slightly elevated when compared to the control.
Figure 5 and its results. You do see a 50% reduction too for PPARa with respect to IL-6 and a reduction in all for IL1B. Though not significant ALL the PPARs do show a reduction. You can say something about that. Maybe good to suggest a bit more data? The “large” SDs of the QPCR results in no significant data. So extra experiment may show your effects to be present.
Major: ref 19 should be in the introduction. As they already did the trick in mice you now show this in a second model. Should be in there. They already suggested this role you investigate.
Major: I miss the discussion of the IL-1 and 6 in the discussion.
Line 173: “The anti-inflammatory effect” <<too strong, to unspecific>>… mention the effect (reduction of NFKB and TNFa indicate anti-inflammatory effects as observed in all…..
Author Response
First of all an interesting paper of the role of PPARs on corneal damage healing. Of course, there are a few things to consider:
Line 30: Check the usage of PPAR as multiple proteins or as “one”.
“peroxisome proliferator-activated receptor (PPAR) is a nuclear receptor”
Or is it:
peroxisome proliferator-activated receptors (PPARs) are a group of nuclear receptors
then the second sentence:
“The PPAR has three isoforms” would be: PPARs comprise of three main isoforms.
Note “main” is probably better as splice variants are also isoforms and there are several for each PPAR.
→We corrected the sentence according to your comment. (line 30-32)
see line 40: subtypes of PPAR s are…
Check this throughout the manuscript.
→The word "PPAR" was changed to "PPARs" throughout the manuscript.
Line 33: I am sure that PPARa and PPARy are not recognized as drugs! I suppose you mean: PPARa and PPARy <<agonist>>. Please check this. Note sentence is also really long.
→You are right. The sentences were corrected. (line 34)
Line 34: Add a dot after the “[5.6]” and start a new sentence “Several…..”
→Corrections were made. (line 35)
Line 37: here you mention “PPARa and y” while before “PPARa and PPARy” was used. Please be consistent and use the PPAR before the y or b/d all the time. (like for example what is done in line 90)
→We corrected the sentence according to your comment. (line 39)
Line 36-39: good that you mention your own research. Its unclear however whether the second inflammatory response was found in the corneal wound model. Please elaborate. Note maybe good to include others as well as PPARs are well known to suppress inflammation.
→We added descriptions regarding PPAR and inflammation in line 36-39.
Line 39-40: “Each PPAR is localized differently in the eye, and differences in their localization are thought to be involved in PPAR function.” References are missing and this info is useless now please elaborate how its then differentially expressed.
→We added a reference (line 41) and corrected the sentence. (line 40-42)
Next line you state that they are ALL present in the corneal epithelium. How are the expression levels? Just one reference? Directly isolated or tissue culture? Seems this is a study from one of the authors please mention in that case “we found that all subtypes” line 40.
→The author is a member of our lab, so we corrected the sentence according to your comment. (line 42-43) The study in the reference showed that expressions of each PPARs by only immunohistochemical analysis. Each PPARs was expressed in the epithelial basement cells of normal cornea and re-epithelialized cornea after alkali burn. We couldn’t find other reports that investigated the localization of PPARs in the cornea.
Line 56-58: “In each…. groups” is bit overdone. remove this.
→We deleted that sentence.
Just change line 51-52: “The ratio of the epithelial defected area to the original defected area at each time point was significantly lower for the PPAR treatments after 12h (p<0.05) and even more pronounced at 24h (p<0.01) (Figure 1b).”
<<note: this makes the sentence more useful instead of “materials and methods”.>>
→We corrected the manuscript according to your comment. (line 56-58)
Major point:
Figure 2 e: I only see an increase in PPARb/d not for the others! Please then don’t pretend with a “greater increase” as there is no increase for the other PPARs. Change this. Ki67 is unchanged for PPARa and PPARy though interestingly increased for PPARb/d
Also change this in the discussion.
→Your comment is exactly right. We deleted the incorrect sentence you pointed out and added result of the count of KI67-positive cells. (line 82-84)
Line 69: for the other two PPARs you could better state: “seemed more enhanced”.
→The above sentence was deleted due to the addition of the count results of KI67-positive cells.
Maybe even better to start with the mRNA as that result is clear and supported by the immunostaining. As normally you present the clearest typical pictures. Therefore, for me only the PPARb/d reflect an increase in protein supporting your mRNA increase the other two might be slightly elevated when compared to the control.
→ Based on the comment of another reviewer, we have added the cell count of Ki67 immunostaining. (line 82-83,88). In this case, still should we change order of immunostaining and mRNA?. If we should change, we will revise.
Figure 5 and its results. You do see a 50% reduction too for PPARa with respect to IL-6 and a reduction in all for IL1B. Though not significant ALL the PPARs do show a reduction. You can say something about that. Maybe good to suggest a bit more data? The “large” SDs of the QPCR results in no significant data. So extra experiment may show your effects to be present.
→We performed PCR analysis and new data were added. There was a significant difference in IL-1b and Il-6 expression between all PPARs and controls. (line137,141-142)
Major: ref 19 should be in the introduction. As they already did the trick in mice you now show this in a second model. Should be in there. They already suggested this role you investigate.
→We moved the ref 19 as ref 17 to the introduction. (line 44)
Major: I miss the discussion of the IL-1 and 6 in the discussion.
→We added discussion of the IL-1 and 6. (line 185)
Line 173: “The anti-inflammatory effect” <<too strong, to unspecific>>… mention the effect (reduction of NFKB and TNFa indicate anti-inflammatory effects as observed in all…..
→We corrected the sentence according to your comment. (line 193)
Reviewer 2 Report
In this study the effects of some selective PPAR agonists on corneal epithelial wound healing were investigated. Overall, the research sounds well and the organization of the manuscript appears well done. The experiments described were carried out with a good methodologic approach.
However, I find that this manuscript requires a revision before publication in this journal, as follows:
1) the presence of different PPAR isotypes in the corneal epithelium is not analyzed by authors. I think it is important, to understand the different behavior played by selective alpha, gamma or delta PPAR agonists. The expression of different isotypes should be measured.
2) The effects analyzed by authors in three PPAR groups should be related to PPAR activation, but there is not a demonstration of this. The authors should perform experiments with combinations of agonists and antagonists for each PPAR isotype, so giving evidence that the observed effect is PPAR-mediated. This is a general comment to be extended to all experiments described in the manuscript.
About this point, the authors state (page 8, lines 145-148)" In the present study, PPARβ/δ ligand increased the expression of Ki67, suggesting that activation of PPARβ/δ promotes wound healing by improving proliferative ability. However, contrary to the present results, Gu et al. reported that a PPARβ/δ agonist suppressed Ki67, and a PPARβ/δ antagonist promoted Ki67 in the rat phototherapeutic keratectomy model", supporting the need for studies combining PPAR agonists and antagonists.
3) I suggest to change the title into "Effects of selective ...agonists on corneal...., that I find more appropriate.
Author Response
In this study the effects of some selective PPAR agonists on corneal epithelial wound healing were investigated. Overall, the research sounds well and the organization of the manuscript appears well done. The experiments described were carried out with a good methodologic approach.
However, I find that this manuscript requires a revision before publication in this journal, as follows:
1) the presence of different PPAR isotypes in the corneal epithelium is not analyzed by authors. I think it is important, to understand the different behavior played by selective alpha, gamma or delta PPAR agonists. The expression of different isotypes should be measured.
→mRNA expressions of all PPAR subtypes were analyzed. (line 53-54, 63-66)
2) The effects analyzed by authors in three PPAR groups should be related to PPAR activation, but there is not a demonstration of this. The authors should perform experiments with combinations of agonists and antagonists for each PPAR isotype, so giving evidence that the observed effect is PPAR-mediated. This is a general comment to be extended to all experiments described in the manuscript.
About this point, the authors state (page 8, lines 145-148)" In the present study, PPARβ/δ ligand increased the expression of Ki67, suggesting that activation of PPARβ/δ promotes wound healing by improving proliferative ability. However, contrary to the present results, Gu et al. reported that a PPARβ/δ agonist suppressed Ki67, and a PPARβ/δ antagonist promoted Ki67 in the rat phototherapeutic keratectomy model", supporting the need for studies combining PPAR agonists and antagonists.
→Thank you for your valuable comments that are very helpful. We now plan investigating effects of PPAR antagonists in the near future. 
3) I suggest to change the title into "Effects of selective ...agonists on corneal...., that I find more appropriate.
→We corrected the title according to your comment. (line 2)
Reviewer 3 Report
The effects of the activation of all three PPAR isotypes on corneal epithelial wound healing are presented in this manuscript. The model used is rat corneal alkali burn. The authors found that the three PPAR isotypes accelerate healing via an increase in epithelial cell proliferation and a reduction of the inflammatory response.
These results are of great interest but the manuscript presents weaknesses that must be corrected to make the paper more attractive and stronger. In the points below, suggestions are made to reach this goal.
1) Results of Figure 1. It is not clear what was measured in the pictures of Figure 1a to obtain the results of Figure 1b. The areas measured should by identified by tracing over the pictures. Otherwise, these results are not accessible to nonspecialists.
2) In Figure 1, healing was measured after 12 and 24-hour treatments. From Figure 2 onwards, the measurements were made after 6 hours of treatment. This change should be justified in the text.
3) Results of Figure 2a-d. Immunostaining of Ki67. The pictures only provide a qualitative presentation of the observations. The results should be quantified by image analysis to provide more objective quantitative results. At the level of the mRNA, an increase of Ki67 is observed only after treatment with the PPARβ/δ ligand, the authors say that, at the protein level, all three PPAR treatments caused an increase. This underscores the necessity of quantification of the staining. If the results are confirmed, the difference between Ki67 protein and mRNA levels should be discussed briefly.
4) Results of Figure 3. Immunostaining reveals an important reduction in NF-κB positive nuclei by the treatments, while the reduction in NF-κB mRNA is much more modest. This is not properly presented and discussed.
5) Results of Figure 4. Again there is a marked discrepancy between I-κBα mRNA levels and I-κBα positive nuclei. Please comment. The pictures of Figure 4g are of insufficient quality and there is no quantification of the results. In general, how has the specificity of the antibodies been tested?
6) Results of Figure 5. The treatments lead to a reduction in the mRNA levels of TNFα and IL-6. This is not seen for IL-1β. It can be suspected that significance was not reached with IL-1β in all three treatments because of the high variability of the results observed with the vehicle. A critical reconsideration of the IL-1β results is warranted.
7) Based on the results obtained with PPAR isotype-specific ligands and in the interest of potential therapeutic advances, it is surprising that dual- and pan-PPAR agonists have not been tested, such as the PPARα/γ dual agonist saroglitazar, the PPARα/δ dual agonist elafibranor, or the PPAR panagonist lanifibranor. This would make this manuscript much stronger and interesting.
Minor points
8) Line 31: replace isoforms by subtypes or isotypes.
9) Lines 33-34: PPARs are not drugs, but drug targets.
10) The group of Xiomeng Wang in Singapore has also studied the role of PPARβ/δ in retinal angiogenesis; see PMID: 32575793. The authors may want to cite this paper, too.
In conclusion, this paper is potentially very interesting but remains somewhat superficial at this stage.
Author Response
The effects of the activation of all three PPAR isotypes on corneal epithelial wound healing are presented in this manuscript. The model used is rat corneal alkali burn. The authors found that the three PPAR isotypes accelerate healing via an increase in epithelial cell proliferation and a reduction of the inflammatory response.
These results are of great interest but the manuscript presents weaknesses that must be corrected to make the paper more attractive and stronger. In the points below, suggestions are made to reach this goal.
- Results of Figure 1. It is not clear what was measured in the pictures of Figure 1a to obtain
the results of Figure 1b. The areas measured should by identified by tracing over the pictures. Otherwise, these results are not accessible to non specialists.
→We added a supplemental figure.
- In Figure 1, healing was measured after 12 and 24-hour treatments. From Figure 2 onwards, the measurements were made after 6 hours of treatment. This change should be justified in the text.
→We added the sentence according to your comment. (line 79-81)
- Results of Figure 2a-d. Immunostaining of Ki67. The pictures only provide a qualitative presentation of the observations. The results should be quantified by image analysis to provide more objective quantitative results. At the level of the mRNA, an increase of Ki67 is observed only after treatment with the PPARβ/δ ligand, the authors say that, at the protein level, all three PPAR treatments caused an increase. This underscores the necessity of quantification of the staining. If the results are confirmed, the difference between Ki67 protein and mRNA levels should be discussed briefly.
→We added the description of cell count of Ki67 staining. (line 82-84, 89-91)
- Results of Figure 3. Immunostaining reveals an important reduction in NF-κB positive nuclei by the treatments, while the reduction in NF-κB mRNA is much more modest. This is not properly presented and discussed.
→We added the discussion according to your comment.(line 170-175)
- Results of Figure 4. Again there is a marked discrepancy between I-κBα mRNA levels and I-κBα positive nuclei. Please comment. The pictures of Figure 4g are of insufficient quality and there is no quantification of the results. In general, how has the specificity of the antibodies been tested?
→Regarding I-kB, there was some variation in positive cells in the immunostaining results. In addition, as you pointed out, we think that PCR showed increased expression of I-kB mRNA of vehicle and PPARγ that dissociated from the staining results. We will endeavor to summarize and report the results in the next experiment (cell count, etc.) without concluding that the cause was individual differences due to animal experiments and the degree of dyeing of DAB. Frozen sections 6 hours after treatment were difficult (because epithelial cells remain deficient) and we tried many times, but only this image was obtained. Thank you for instructing us on new issues.
- Results of Figure 5. The treatments lead to a reduction in the mRNA levels of TNFα and IL-6. This is not seen for IL-1β. It can be suspected that significance was not reached with IL-1β in all three treatments because of the high variability of the results observed with the vehicle. A critical reconsideration of the IL-1β results is warranted.
→We appreciate the reviewer’s vauable suggestion. We performed PCR analysis and new data were added. There was a significant difference in IL-1b and Il-6 expression between all PPARs and controls. (line137,141-142)
- Based on the results obtained with PPAR isotype-specific ligands and in the interest of potential therapeutic advances, it is surprising that dual- and pan-PPAR agonists have not been tested, such as the PPARα/γ dual agonist saroglitazar, the PPARα/δ dual agonist elafibranor, or the PPAR panagonist lanifibranor. This would make this manuscript much stronger and interesting.
→We agree to the the reviewer’s comment. Actually we have been doing research regarding dual agonists and interesting results were obtained, however, data analysis needs some time. We hope a paper can be published in the near future.
Minor points
8) Line 31: replace isoforms by subtypes or isotypes.
→The word were corrected according to your suggestion. (line 32)
9) Lines 33-34: PPARs are not drugs, but drug targets.
→The word "agonist" was added. (line 34)
10) The group of Xiomeng Wang in Singapore has also studied the role of PPARβ/δ in retinal angiogenesis; see PMID: 32575793. The authors may want to cite this paper, too.
→Thank you for introducing a useful paper. We added it to the references. (line 147)
In conclusion, this paper is potentially very interesting but remains somewhat superficial at this stage.
Round 2
Reviewer 2 Report
In the title please correct agonists instead of agonist " Effects of selective Peroxisome Proliferator Activated Receptor agonists on corneal epithelial wound healing "
The authors corrected PPARs along the manuscript, bur when it is used as an adjective, it requires the singular form. For example, in the first sentence of abstract, "The effects of each subtype of peroxisome proliferator activated receptors (PPARs) agonist (α, β/δ, γ)" should be changed in "The effects of subtype-selective Peroxisome Proliferator Activated Receptor (PPAR) agonists (α, β/δ, γ)....". This is a general rule to respect in all the manuscript.
Author Response
In the title please correct agonists instead of agonist " Effects of selective Peroxisome Proliferator Activated Receptor agonists on corneal epithelial wound healing "
The authors corrected PPARs along the manuscript, bur when it is used as an adjective, it requires the singular form. For example, in the first sentence of abstract, "The effects of each subtype of peroxisome proliferator activated receptors (PPARs) agonist (α, β/δ, γ)" should be changed in "The effects of subtype-selective Peroxisome Proliferator Activated Receptor (PPAR) agonists (α, β/δ, γ)....". This is a general rule to respect in all the manuscript.
→You are right. We corrected according to your comment throughout the manuscript.
Reviewer 3 Report
The revised version provided by the authors is stronger than the original version, but not all comments have been met.
Most importantly, results from dual- and pan-agonist treatments should be included, even if some time is needed to generate the results.
The English language in the revised segments of the paper should be checked.
Author Response
The revised version provided by the authors is stronger than the original version, but not all comments have been met.
Most importantly, results from dual- and pan-agonist treatments should be included, even if some time is needed to generate the results.
The English language in the revised segments of the paper should be checked.
→
Thank you for your comment.
Additional experiment will take a lot of time, because there are restrictions on animal imports and animal experiments due to COVIT-19. Furthermore, another member of my team is considering submitting an article about your request. We will do our best to make further reports from our team in the future. Would you please accept the existing data this time?
We finished the English proofreading.
Round 3
Reviewer 3 Report
The authors did not include data about dual- and pan-PPAR agonists arguing that this will be a different paper and that they face animal experiment restriction.
It is not very satisfying, but I can live with it.